# miR-1 as a Key Epigenetic Regulator in Early Differentiation of Cardiac Sinoatrial Region

**DOI:** 10.3390/ijms25126608

**Published:** 2024-06-15

**Authors:** Carlos García-Padilla, Estefanía Lozano-Velasco, Virginio García-López, Amelia Aránega, Diego Franco, Virginio García-Martínez, Carmen López-Sánchez

**Affiliations:** 1Department of Human Anatomy and Embryology, Faculty of Medicine and Health Sciences, Institute of Molecular Pathology Biomarkers, University of Extremadura, 06006 Badajoz, Spain; carlosgp@unex.es (C.G.-P.); evelasco@ujaen.es (E.L.-V.); garcialopez@unex.es (V.G.-L.); virginio@unex.es (V.G.-M.); 2Department of Experimental Biology, University of Jaen, 23071 Jaen, Spain; aaranega@ujaen.es (A.A.); dfranco@ujaen.es (D.F.); 3Medina Foundation, 18016 Granada, Spain; 4Department of Medical and Surgical Therapeutics, Pharmacology Area, Faculty of Medicine and Health Sciences, University of Extremadura, 06006 Badajoz, Spain

**Keywords:** miR-1, epigenetic regulator, cardiac sinoatrial region, molecular mechanisms, *Hdac4*, *Calm1*/*Calmodulin*, Erk2/*Mapk1*, MEF2C activity, retinoic acid signaling, cardiac development

## Abstract

A large diversity of epigenetic factors, such as microRNAs and histones modifications, are known to be capable of regulating gene expression without altering DNA sequence itself. In particular, miR-1 is considered the first essential microRNA in cardiac development. In this study, miR-1 potential role in early cardiac chamber differentiation was analyzed through specific signaling pathways. For this, we performed in chick embryos functional experiments by means of miR-1 microinjections into the posterior cardiac precursors—of both primitive endocardial tubes—committed to sinoatrial region fates. Subsequently, embryos were subjected to whole mount in situ hybridization, immunohistochemistry and RT-qPCR analysis. As a relevant novelty, our results revealed that miR-1 increased *Amhc1*, *Tbx5* and *Gata4*, while this microRNA diminished *Mef2c* and *Cripto* expressions during early differentiation of the cardiac sinoatrial region. Furthermore, we observed in this developmental context that miR-1 upregulated *CrabpII* and *Rarß* and downregulated *CrabpI*, which are three crucial factors in the retinoic acid signaling pathway. Interestingly, we also noticed that miR-1 directly interacted with *Hdac4* and *Calm1*/*Calmodulin*, as well as with *Erk2*/*Mapk1*, which are three key factors actively involved in *Mef2c* regulation. Our study shows, for the first time, a key role of miR-1 as an epigenetic regulator in the early differentiation of the cardiac sinoatrial region through orchestrating opposite actions between retinoic acid and *Mef2c*, fundamental to properly assign cardiac cells to their respective heart chambers. A better understanding of those molecular mechanisms modulated by miR-1 will definitely help in fields applied to therapy and cardiac regeneration and repair.

## 1. Introduction

During early embryogenesis, the cardiogenic mesoderm organizes—at both sides of the embryo—the primitive endocardial tubes, which fuse to form the primitive heart tube in the midline [1,2,3,4]. Subsequently, the posterior cardiac tube segment differentiates into the cardiac sinoatrial region, including the *atrium* and inflow tract/*sinus venosus* [5,6]. Concomitantly, the anterior cardiac tube segment contributes to ventricular and outflow tract differentiation [7,8]. Previous studies have revealed that this segmentation process is regulated by several cardiac transcription factors, including *Tbx5*, *Gata4* and *Mef2c*, among others, which activate the expression of early cardiac structural genes, such as myosin heavy-chain (*MHC*) and myosin light-chain (*MLC*) genes [9,10,11,12,13]. It is also well known that the phenotype of cardiac sinoatrial region is influenced by retinoic acid (RA) signaling, mediating the expression pattern of early molecular markers such as *Tbx5*, *Gata4* and *AMHC1* (atrial myosin heavy chain), which together determine the posterior heart tube segment, leading to venous and atrial cell fate [14,15,16]. In fact, exclusion of RA from ventricular precursors is essential for correct specification of the ventricles [17]. Furthermore, RA synthesis is largely controlled by retinaldehyde dehydrogenase 2 (*Raldh2*/*Aldh1a2*) [18,19], in such a way that *Raldh2*^−/−^ knockout mice present reduced *Tbx5* and *Gata4* expression and severely impaired cardiac sinoatrial region development, while *Mef2c* expression, which induces ventricular cardiomyocyte differentiation [20], was unaltered [21]. Along with the above, *Mef2c*^−/−^ knockout mice displayed ectopic *Tbx5* expression, suggesting that the mutant ventricle acquires atrial-specific characteristics and, therefore, *Mef2c* could act directly on cells and make them nonresponsive to RA inflow differentiation signaling [22].

Several authors have also reported that a large diversity of epigenetic processes, including the expression of non-coding RNAs (especially microRNAs), histone modifications and DNA methylation, are capable of regulating gene expression by influencing transcription or inhibiting translation through several functional overlapping and cross-talking mechanisms [23,24]. In particular, microRNAs are considered as a post-transcriptional type of epigenetic regulatory processes, since the degree of complementarity microRNA/target mRNA determines its final outcome, either by degradation or translational inhibition [24,25]. Specifically, miR-1 has been the first essential microRNA identified in cardiac development, with a specific expression pattern in cardiac muscle [26,27,28,29,30,31]. Interestingly, experimental overexpression of miR-1 during heart development results in defective ventricular myocyte proliferation, accompanied by hypoplasia of the cardiac ventricular conduction system [32]. To date, the specific effects of miR-1 during early differentiation on cardiac sinoatrial region are unknown. The previous study that analyzed the functional role of miR-1, which was performed in murine embryonic stem cells, was focused on the development and function of sinoatrial cells, corresponding specifically to the sinoatrial node, which determines the cardiac pacemaker [33].

Furthermore, it has been proposed that miR-1 modulates cardiomyocyte growth responses by negatively regulating *Calm1*/*Calmodulin*, a miR-1 target gene [34]. In addition, miR-1 regulates human cardiomyocyte progenitor cell differentiation via the repression of histone deacetylase 4 (*Hdac4*), also identified as a miR-1 target gene, altering histone-DNA-binding activity [35]. Nevertheless, *Hdac4* can also interact with non-histones by covering the promoter region of a gene, thus inhibiting its expression [36,37,38]. For instance, *Hdac4* can interact with *Mef2c* by repressing its transcriptional activity, which is required for muscle cell differentiation [39]. Additionally, as described in cardiac hypertrophy, it has been reported that the activation of *Mef2c* transcription is mediated both by CALM1 and ERK1/2 -extracellular signal regulated kinases 1 and 2, a signal transducer in the family of mitogen activated protein kinase, MAPK- [40,41,42,43,44]. In this sense, it has been recently reported, in a zebrafish model, that ERK1/2 signaling is required to promote ventricular chamber differentiation [45].

Given the complexity of those molecular mechanisms involved in cardiac chamber formation, this experimental study aims to analyze miR-1 potential functions in early differentiation of the cardiac sinoatrial region. By means of gain- and loss-of-function experiments, we show that miR-1 plays a dual role by modulating both MEF2C activity and RA signaling pathway, placed as a key epigenetic regulator of appropriate cardiac cell specification, thus promoting the initial differentiation of the cardiac sinoatrial region.

## 2. Results

### 2.1. miR-1 Regulates Specific Gene Expressions during Early Differentiation of the Cardiac Sinoatrial Region

In this research work, we carried out miR-1 gain- and loss-of-function experiments by means of respective microinjections of premiR-1 and antimiR-1 into the posterior cardiac precursors—in primitive endocardial tubes—committed to sinoatrial region fates (Figure 1). Noticeably, in experimental embryos, we observed morphological alterations based on an expansion of the posterior domain of the heart after miR-1 administration, concomitantly with a reduced ventricle region (Figure 1A,B). This phenotype is similar to that observed following RA treatment and after the deletion of *Mef2c* [22,46]. Additionally, in an experimental group, cardiac *asa* were dissected and subsequently subjected to RNA extraction and RT-qPCR analysis under each experimental condition. Also, we performed RT-qPCR analyses, thus validating the miR-1 gain- and loss-of-function results with respect to the control on cardiac *asa* and fluorescence visualization of the dissected cardiac *asa* (Figure 1C).

Since, in our experimental study, the overexpression of miR-1 gave rise to an expansion of the posterior domain of the heart, we also analyzed the effects of miR-1 on RA target genes, including *Amhc1*, *Tbx5* and *Gata4*, as well as *Mef2c* expression during early differentiation of the cardiac sinoatrial region. As illustrated in Figure 2, Appendix A, our results showed that miR-1 led to increased *Amhc1*, *Tbx5* and *Gata4* expressions. Also, RT-qPCR analyses support these findings (Figure 2A–C). Interestingly, miR-1 inhibited mRNA and protein levels of *Mef2c* gene in the cardiac sinoatrial region (Figure 2D). Supporting all the above data, our miR-1 loss-of-function experiments showed an opposite effect.

Additionally, in this work, we observed that microinjections of RA—into the same areas mentioned above—positively mediated the expression of *Gata4* (Appendix A). This result coincides with the fact that RA increases *Tbx5* and *Amhc1* expression, as we previously reported [46], thus revealing that miR-1 mimics the effects observed after RA administration.

### 2.2. miR-1 Modulates RA Signaling Pathway during Early Differentiation of the Cardiac Sinoatrial Region

Since miR-1 modulates RA target gene expression during differentiation of cardiac sinoatrial region, through RT-qPCR, we analyzed RNAs from the dissected cardiac *asa* of embryos microinjected either with premiR-1 or antimiR-1, in order to explore the relationship between miR-1 and those factors involved in the RA signaling pathway (Figure 3).

Our findings revealed that miR-1 did not modify the expression levels of *Raldh2* (Figure 3A), a RA synthesis modulator [47]. We also observed that miR-1 was able to increase the endogenous expression levels of *CrabpII* (Figure 3B), a RA target gene that regulates RA delivery to its nuclear receptors (RARs) [48,49]. In addition, our results showed that miR-1 ked to reduced levels of *CrabpI* (Figure 3C), which functions as a sequester of RA to facilitate its catabolism [50], thus supporting that miR-1 may upregulate RA signaling. In line with the above results, we obtained the opposite effects through antimiR-1 administration (Figure 3).

With respect to the nuclear retinoic acid receptors, it is noteworthy that *Rarß* is a RA target gene essential for cardiovascular development [51,52,53,54]. Using immunohistochemistry (IMH), in this study, we included a detailed analysis of this cardiac gene. In a control group of embryos, we identified *Rarß* distribution starting in both primitive endocardial tubes, followed by the primitive cardiac tube and continuing in the early cardiac looping formation, presenting a location in sinoatrial region (Appendix A). Our miR-1 gain-of-function experiments led to an increased expression of *Rarß* and its protein levels (Figure 4, Appendix A). Supporting these results, we obtained the opposite effect after our miR-1 loss-of-function experiments (Figure 4, right column, and RT-qPCRs; Appendix A). On the other hand, expression modifications of nuclear retinoic acid receptors *Rarα*, *Rxrα*, *Rarγ* and *Rxrγ* were not detected after miR-1 experimental assays (Figure 5).

### 2.3. miR-1 Modulates Hdac4, Calm1/Calmodulin and Erk2/Mapk1 during Early Differentiation of the Cardiac Sinoatrial Region

Given that our experiments revealed that miR-1 decreases both mRNA and protein levels of *Mef2c*, which is not a miR-1 target gene, we searched for pivotal miR-1 targets genes as potential *Mef2c* regulators during differentiation of cardiac sinoatrial region. Using bioinformatics analyses through Target-Scan software, we identified *Hdac4*, *Calm1*/*Calmodulin* and *Erk2*/*Mapk1* as putative targets of miR-1 (Appendix A), suggesting that it might exert control on their post-transcriptional regulation. Our luciferase assays showed that luciferase signals from plasmids harboring 3′UTRs of each gene were reduced with respect to the control, proving that miR-1 recognizes and directly binds to *Hdac4*, *Calm1* and *Erk2*/*Mapk1* 3′UTRs, thus triggering their mRNA degradation. Supporting these findings, and showing the specificity to the predicted sites, the reduced activity by miR-1 was not observed in *Hdac4*, *Calm1* or *Erk2* 3′UTR reporter vectors containing the mutated sequences of the predicted miR-1 binding sites, respectively (Figure 6).

In line with these results, by means of in situ hybridization (ISH), we analyzed and identified miR-1 expression patterns in early chick embryos, starting in both primitive endocardial tubes, followed by the primitive cardiac tube and continuing in the early cardiac looping formation. By means of IMH, we observed a similar distribution of HDAC4, CALM1 and phospho-ERK2 (pERK2) in the sinoatrial region during the stages under analyses. Interestingly, our comparative analysis demonstrated a complementary expression pattern among them, suggesting a plausible regulatory link during early posterior cardiac tube differentiation. Noticeably, this pattern coincided with the cardiac distribution of MEF2C (Figure 7).

As a clear demonstration of miR-1 functional role on *Mef2c* regulators, our gain-of-function experiments showed that miR-1 repressed both mRNA and protein levels of *Hdca4*, *Calm1* and *Erk2* in the sinoatrial region (Figure 8, Appendix A). In agreement with these results, the opposite effects were observed after antimiR-1 administration (Figure 8, right column, and RT-qPCRs). Therefore, our findings indicate that miR-1 modulates *Mef2c* expression through the repression of *HDAC4*, *Calm1* and *Erk2* during early differentiation of the cardiac sinoatrial region.

Finally, our results showed that miR-1 led to reduced levels of *Cripto*, obtaining the opposite effect after our miR-1 loss-of-function experiments (Figure 9). Since *Cripto*/*Tdgf1*, a direct transcriptional target of MEF2C [55], shows a coincident expression pattern with *Mef2c* during early cardiac development restricted to sinoatrial region [56,57], our findings support miR-1 modulation activity on MEF2C signaling during the differentiation of the sinoatrial region.

## 3. Discussion

In this work, we have obtained original findings about numerous molecular factors involved in the posterior cardiac tube segment differentiation. In this sense, this is the first time there is clear evidence about the function of miR-1 modulating MEF2C activity and RA signaling pathway during early cardiac chamber differentiation. Our results revealed (Figure 10) that miR-1 administration into the posterior cardiac precursors in the primitive endocardial tubes gave rise to high expression levels of *Amhc1*, *Tbx5* and *Gata4*, which have previously demonstrated to be RA target genes, essential for the early differentiation of the cardiac sinoatrial region [16,46,58]. Significantly, the expression of *Mef2c*, a ventricular cardiomyocyte inductor [20,22], was diminished after miR-1 administration. Moreover, *Cripto*—a direct transcriptional target of MEF2C [55]—also was diminished after our miR-1 gain-of-function experiments. Additionally, morphological alterations were detected after miR-1 overexpression, characterized by an enlarged posterior domain of the heart and a reduced ventricle region. All the above results show that miR-1 mimics the effects previously associated to highlighted RA levels [14,15,16,17,18,19,21], indicating that miR-1 actively interferes with one or more factors involved in the RA signaling pathway. In this line, our results revealed that miR-1 upregulated *CrabpII* and *Rarß*, while it downregulated *CrabpI*, three crucial factors in RA signaling pathway. Interestingly, this microRNA also mimicked the results obtained after the deletion of *Mef2c* [20,22], indicating that miR-1 clearly modulates *Mef2c* transcription regulators. In addition to the above data, our results showed that: (i) miR-1 recognized and directly bonded to *Erk2*/*Mapk1*, *Hdac4* and *Calm1*, the last two also previously reported as miR-1 target genes in culture cardiac cells [34,35]; (ii) this microRNA presented a complementary expression pattern with these three target genes during early posterior cardiac tube differentiation; and (iii) it repressed both mRNA and the protein levels of *Hdac4*, *Calm1* and *Erk2*/*Mapk1* in the sinoatrial region. Our findings support the fact that miR-1 actively interacts with these three potential molecular factors and that these factors are involved both in *Mef2c* regulation and RA signaling pathway during early differentiation of the cardiac sinoatrial region (Figure 10).

### 3.1. miR-1 Modulates MEF2C Activity

Our results showed that miR-1 diminished both mRNA and the protein levels of *Mef2c* in the posterior domain of the developing heart. It has been proposed that *Mef2c* promotes outflow-specific differentiation and directs cells in a nonresponsive manner to the inflow-specifying actions of RA [22]. Since *Mef2c* is not a miR-1 target gene, we analyzed the effects of miR-1 on *Mef2c* modulators and observed that this microRNA repressed both mRNA and the protein levels of cytoplasmatic *Hdac4* (*cHdac4*), *Calm1* and *Erk2*/*Mapk1* (Figure 11). In this context, it is particularly known that nuclear *HDAC4* (*nHDAC4*) directly binds to *Mef2c* and blocks its activity [36,39,59]. Through pharmacological inhibition of *Hdac4* activity by trichostatin A (TSA) [60], our experimental model (Appendix A) revealed that *Mef2c* expression presented a dramatic increase, proving the importance of histone deacetylation in this transcription factor regulation. It is also known that *Calm1*—through CaMKII induction—leads to the phosphorylation of nHDAC4 and its translocation to the cytoplasm, thus promoting *Mef2c* activation [40,41,42,44,61]. Therefore, in our model (Figure 11), given that low levels of Calm1—-miR-1-induced—did not allow the export of nHDAC4 from the nucleus to the cytoplasm, miR-1 would increase *Mef2c*’s interaction with nHDAC4, which would inhibit *Mef2c* expression. It has also been reported that nHDAC4, together with the nuclear corepressor NCoR1/SMRT, synergize to inhibit *Mef2c* activity [62,63]. Furthermore, NCoR1/SMRT actively translocates cHDAC4 from the cytoplasm into the nucleus and prevents its nuclear export [63,64,65]. The above data support the fact that there are opposite effects of CALM1 and NCoR1/SMRT on nHDAC4 modulated by miR-

On the other hand, it is known that NCoR1/SMRT phosphorylation by ERK2 may destabilize the association between *Mef2c* and nHDAC4, thus increasing *Mef2c* expression [63,66,67]. It has also been reported that *Mef2c* may be enhanced by ERK1/2 through the upregulation of coactivator P300/CBP-associated factor (PCAF), which inherently presents histone acetyltransferases (HAT) activity, thus increasing both the expression and function of *Mef2c* [43]. Our results showed that miR-1 directly recognized and repressed *Erk2*/*Mapk1* expression (Figure 9), and consequently, this microRNA diminished MEF2C activity; these results are supported by the above data. In this line, it has been observed during zebrafish development that excessive levels of ERK1/2 in atrial location cause the ectopic expression of ventricular specific genes [45], an effect that also supports our model, proposing miR-1 as a key regulator of atrial differentiation by repressing *Erk1*/*2* expression.

Based on all the molecular mechanisms mentioned above, we propose in our model (Figure 11) that, in these early stages of cardiac development, miR-1 plays a crucial role to repress *Mef2c* through *Erk2*/*Mapk1*, *Hdac4* and *Calm1* modulation.

Noticeably, our results showed that miR-1 diminished the expression level of *Cripto*. It is known that MEF2C is required for *Cripto* enhancer activity and expression during cardiac development [55]. Also, it has been reported in in vitro models during cardiac differentiation, that high levels of *Cripto* gene expression coincide with low levels of miR-1 *and vice versa*, establishing a crosstalk between miR-1 and *Cripto* during cardiomyogenesis [68]. Furthermore, in mice, *Cripto* is a direct target of miR-1; thus, when miR-1 binds to the 3′UTR *Cripto*, it suppresses its expression and promotes cardiomyocyte specification. Nevertheless, in silico analysis reveals that *Cripto* is not a direct target of miR-1 in chicken, thus suggesting that this interaction could be mediated through MEF2C. It has also been reported that *Cripto* is strongly expressed in undifferentiated cells, and it is rapidly downregulated during retinoic acid-induced differentiation [69]. Therefore, in our model, miR-1 could be able to promote the RA signaling pathway to induce the early differentiation of the cardiac sinoatrial region.

### 3.2. miR-1 Modulates the RA Signaling Pathway

With respect to miR-1 inductive role on the RA signaling pathway during the early differentiation of the cardiac sinoatrial region (Figure 10), we found that this microRNA increased the expression levels of RA target genes *Tbx5*, *Gata4* and *Amhc1*. It is known that RA establishes an epigenetic switch for histone acetylation, mediated by p300/CBP, which inherently presents histone acetyltransferases (HAT) activity to allow transcription of RARs target genes [70,71,72,73,74,75]. We also observed that miR-1 repressed epigenetic repressor *Hdac4*, which has been reported as a negative regulator of RA target genes [76]. In line with the above, we showed in this work that *Tbx5*, *Gata4* and *Amhc1* expressions increased with HDAC4 inhibitor TSA (Appendix A), confirming that RA regulates these cardiac genes through direct modulation of histone deacetylation. Furthermore, previous studies have reported that nHDAC4 is recruited and retained in the nucleus by NCoR1/SMRT, which is associated to RARs in the absence of RA [64,77,78,79]. As a matter of fact, it is known that the presence of RA disrupts the interaction of NCoR1/SMRT with RARs [67,80], thus allowing the transcription of RARs’ target genes (Figure 11). Other studies have described that ERK2/MAPK1 phosphorylates nuclear corepressors N-CoR1/SMRT, reducing the interaction between N-CoR1/SMRT and RARα [66], which would allow us to suggest an increase of RARs target genes transcription. However, our results showed that miR-1 directly suppressed *Erk2*/*Mapk1* and increased RA target genes. Based on these results, we propose that *Erk2*/*Mapk1* is not sufficient to modulate atrial gene expressions, so an RA signal is necessary to disrupt the interaction between N-CoR1/SMRT and RARs [77]. Keeping all the above in mind, our results showed that miR-1 upregulated RA function, thus promoting the expression of its target genes.

Interestingly, in our study, we found that miR-1 increased the expression levels of *CrabpII*, a RA target gene—with an RA response element (RARE)—involved in delivering RA to its nuclear receptors RARs [48,49,81,82]. In addition, we observed that miR-1 increased both mRNA and the proteins of *Rarβ*, a previously defined RA target gene with RARE located in its promoter [51,54,73,80,82]. Moreover, by means of IMH, we carried out a detailed dynamic analysis of RARβ distribution, which showed a specific location in the cardiac sinoatrial region. The above findings, together with the fact that RARβ repression is linked to cardiac abnormalities [52], support miR-1 active role on RA signaling pathway through *Rarβ*. Nevertheless, the levels of nuclear retinoic acid receptors *Rarα*, *Rxrα*, *Rarγ* and *Rxrγ* did not show significant changes after our miR-1 experimental assays, proving that these receptors are not targeted either by miR-1 or RA. It is noteworthy, based on our results, that miR-1 directly suppresses *Caml1*, a protein that also acts as an inhibitor of RARα activity by means of CaMKII induction, which mediates RARα phosphorylation and enhances the interaction between RARα and N-CoR1/SMRT, and subsequently supresses RARα target genes transcription [83]. In our proposed model (Figure 11), miR-1 repressed CALM1’s capability to inhibit RARs’ activity, thus promoting the expression of RA target genes. Supporting these results, previous studies have shown that the blockage of RARα function diminishes *Gata4* transcripts, specifically interfering with the inflow tract formation [84,85].

Additionally, our results show that miR-1 overexpression did not generate significant changes in the expression level of *Raldh2*, a RA synthesis modulator [47,79,86] involved in cardiac sinoatrial region development [18,19]. Supporting our results, previous studies [87] have shown that *Raldh2* expression is not altered in a nuclear corepressor mutant mouse model—SMRTmRID—which is characterized by enhanced transcription of RARs targets.

Noticeably, we observed that miR-1 diminished the expression levels of *CrabpI*, a protein described as a sequester of RA, facilitating its catabolism, limiting RA concentration and regulating the amount of RA that is accessible to nuclear receptors RARs [50,88]. This supports the fact that miR-1 enhances RA activity. Given that *CrabpI* is not a miR-1 target gene, our model (Figure 11) proposes that miR-1 inhibits *Med1*/*Trap220* (a miR-1 target gene identified *through* in silico analysis), which is a *CrabpI* coactivator associated with RA [89]. Therefore, miR-1 would be able to promote RA signals.

Taking into account all the above, our novel experimental model integrates a network of molecular mechanisms modulated by miR-1, which plays a crucial role during early stages of cardiac chamber formation by promoting atrial differentiation together with complementary suppression of ventricular formation. In conclusion, our study reveals, for the first time, a key role of miR-1 as an epigenetic factor modulating RA and *Mef2c* in their opposite actions, which are required to properly assign cells to their respective cardiac chambers. Since miR-1 has also been identified in cardiomyopathy processes [90,91,92,93,94,95], further and deeper understanding of this microRNA as a modulator of molecular mechanisms governing specific signaling pathways could be helpful in therapy and cardiac regeneration and repair.

## 4. Materials and Methods

Experimental protocols with animals were performed in agreement with the Spanish law in application of the EU Guidelines for animal research and conformed to the Guide for the Care and Use of Laboratory Animals, published by the US National Institutes of Health (NIH Publication no.85-23). Approval by the University of Extremadura bioethics board was obtained prior to the initiation of the study.

### 4.1. Early Chick Whole Embryo Culture

Fertilized eggs (Granja Santa Isabel, Córdoba, Spain) were incubated at 37 °C in forced draft humidified incubators. Embryos were staged (HH stages) [96,97] and subjected to early chick (EC) embryo culture [98].

### 4.2. Embryo Microinjections into the Posterior Cardiac Precursors of Both Primitive Endocardial Tubes

Stage HH 8 (3–4 somites) cultured embryos were microinjected (using an Inject + Matic micro injector system) in both primitive endocardial tubes into the posterior cardiac precursors committed to sinoatrial region fates [19,46].

For gain-of-function experiments, two different groups of embryos were microinjected with premiR-1 (Thermo Fisher AM17150–Vacaville, CA, USA) and all-trans retinoic acid (RA; Sigma R2625–St. Louis, MO, USA), respectively. premiR is a precursor miRNA that contains single stem-looped structure to be further processed into mature miRNA, which takes place in the miRNA-induced silencing complex. This, based on the sequence complementary between target mRNA and miRNA, gives rise to translation repression or target RNA degradation [99,100].

Likewise, for loss-of-function experiments, three different group of embryos were microinjected with antimiR-1 (Thermo Fisher AM17000–Vacaville USA), which inhibits the binding of miRNA with its target mRNA; with an inhibitor of RA synthesis, Citral (3,7-dimethyl-2,6-octadienal, Sigma C83007–St. Louis, MO, USA); and with an inhibitor of HDAC, trichostatin A (TSA, Sigma T1952–St. Louis, MO, USA). A working solution containing 2.5 mM CFDA (5-(and-6)-Carboxyfluorescein Diacetate, Thermo Fisher V12883–Vacaville USA) and 1/10 volume of 0.5% (wt/vol) fast green FCF was prepared for both experimental and control (CFDA) embryos. In the experimental embryos, the working solution additionally contained a final concentration of 1μM for microRNAs premiR-1 or antimiR-1 of 10 µg/mL for RA, of 10 mmol/L for Citral, and as well as of 50 nmol/L for TSA.

After 10–12h of additional incubation, embryos were photographed under bright and fluorescent light (Nikon digital, SIGHT DS-U1) and were selected according to the location and extent of the injection. The selected embryos were either fixed in 4% PFA—processed for gene expression (in situ hybridization, ISH) or immunochemistry (IMH) analysis—or cardiac loops were collected for RNA isolation as previously described [46].

### 4.3. Whole-Mount In Situ Hybridization (ISH)

Two different ISH procedures were performed following our previous procedures [101]. One group of control embryos was processed [102] for LNA-ISH using miR-1 LNA-labelled microRNA probe (miRCURY LNA™ Detection probe 5′-DIG and 3′-DIG labelled, Exiqon), while another group of premiR-1, antimiR-1, RA, Citral and control (CFDA) embryos were processed for ISH following standard procedures [103] using antisense-*Tbx5-*, -*Gata4-* and -*Amhc1*-labelled probes [58].

### 4.4. Whole-Mount Immunohistochemestry (IMH)

Experimental and control (CFDA) embryos were subjected to whole mount IMH performed as we previously described [101,104], using polyclonal rabbit antibodies for HDAC4 (1:100, Invitrogen PA5-29103), phospho-ERK1/ERK2 (1:20, Invitrogen 44-680G), RARß (1:100, Invitrogen PA5-33016) and MEF2C (1:50, Proteintech 18290-1-AP), followed by goat anti-rabbit IgG-HRP antibody (1:1000, Upstate, 12-348) and monoclonal mouse antibody for CALM1/Calmodulin (1:20, Invitrogen MA3-918), followed by goat anti-mouse Ig-HRP antibody (1:200, Jackson Immunoresearch Laboratories, West Grove, PA, USA).

### 4.5. RNA Isolation and RT-qPCR

Cardiac loops isolated from experimental and control (CFDA) embryos (Figure 1) were subjected to RT-qPCR analysis following MIQE guidelines [105,106,107]. RNA was extracted and purified using ReliaPrep RNA Cell Miniprep System Kit (Promega) according to the manufacturer’s instructions. For mRNA expression measurements, 0.5 μg of total RNA was used for retro-transcription with Maxima First Strand cDNA Synthesis Kit for RT-qPCR (Thermo Fisher Vicaville USA). Real-time PCR experiments were performed with 2 μL of cDNA, GoTaq qPCR Master Mix (Promega) and corresponding primer sets (Appendix A). For miR-1 expression analyses, 20 ng of total RNA was used for retro-transcription with Universal cDNA Synthesis Kit II (Exiqon), and the resulting cDNA was diluted 1/80. Real-time PCR experiments were performed with 1 μL of diluted cDNA, Go Taq qPCR Master Mix (Promega) as well. All RT-qPCRs were performed using a CFX384TM thermocycler (Bio-Rad) following the manufacture’s recommendations. The relative expression of each gene was calculated using *Gusb* and *Gapdh* as internal controls for mRNA expression analyses and 5S and 6U for miR-1 expression analyses, respectively [108]. Each PCR reaction was carried out in triplicate and repeated in at least three distinct biological samples to obtain representative means.

### 4.6. Analysis In Silico

Target-Scan 8 software was used to perform bioinformatics analysis of the binding sites of miR-1 at 3′UTRs of predicted targets as described before [109,110].

### 4.7. Luciferase Assays and 3T3 Transfection

For luciferase assays, *Hdac4*, *Calm1*/*Calmodulin* and *Erk2*/*Mapk1* 3′ untranslated regions (UTRs) were amplified from mouse genomic DNA and cloned into the pGLuc-Basic vector (New England BioLabs Ipswich England). *Hdac4*, *Calm1* and *Erk2*/*Mapk1* gene 3′ UTRs were amplified from chicken genomic DNA with specific primers bearing HindIII/BamHI restriction sites and cloned into the pGLuc-Basic vector (New England BioLabs). PCR-based site-directed mutagenesis was performed using the Stratagene QuikChange site-directed mutagenesis kit but with the enzymes and buffers from the Bio-Rad iPROOF PCR kit. Primers used for site-directed mutagenesis (Appendix A) introduced mutations into miR-1 seed sequence present in the *Hdac4* 3′ UTR, *Calm1* 3′ UTR and *Erk2*/*Mapk1* 3′ UTR. Independent cotransfection experiments with pre-miRNAs were carried out simultaneously in 3T3 cells with 20 μL of culture medium; luciferase activity was measured 24 h after transfection using a Pierce™ Gaussia Luciferase Flash Assay Kit or a Pierce™ Cypridine Luciferase Flash Assay Kit (Thermo Fisher Vicaville USA). In all cases, transfections were carried out in triplicate.

### 4.8. Statistical Analyses

For statistical analyses of datasets, unpaired Student’s t-tests were used as previously reported [111,112]. Significance levels or *p*-values are stated in each corresponding figure legend. A *p* < 0.05 was considered statistically significant.

### 4.9. Image Analysis

Quantitative analysis of images has been performed using the Image J^®^ software (v.1342).

## Figures and Tables

**Figure 1 ijms-25-06608-f001:**
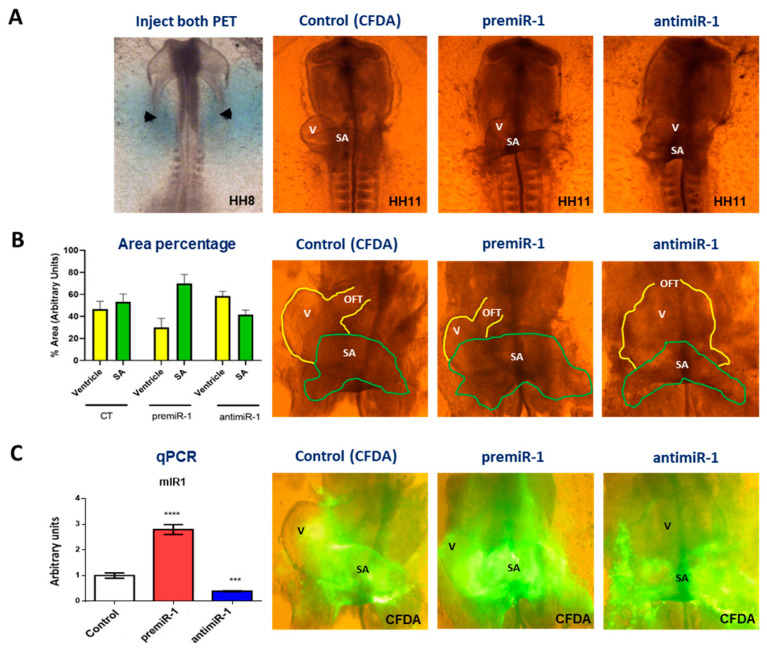
(**A**) Embryo microinjected with CFDA (control), premiR-1 or antimiR-1 at the level of the prospective atrial cells of both primitive endocardial tubes (arrowheads) and light microscopy of control and experimental embryos (HH11) subjected to miR-1 gain- and loss-of-function. Note, in the experimental embryos and dissected cardiac *asa* (HH11), the morphological alterations based on an expansion of the posterior domain of the heart after miR-1 administration, concomitantly with a reduced ventricle region. The opposite effects were observed after antimiR-1 administration. (**B**) Area percentage of structural SA and V after miR-1 gain- and loss-of-function experiments. (**C**) RT-qPCR validating the miR-1 gain- and loss-of-function results respect to control on cardiac *asa* and fluorescence visualization of dissected cardiac *asa*. OFT: outflow tract; V: ventricle; SA: sinoatrial region. Student’s *t*-test: *** *p* < 0.005, **** *p* < 0.0001 with respect to control (CFDA) embryos.

**Figure 2 ijms-25-06608-f002:**
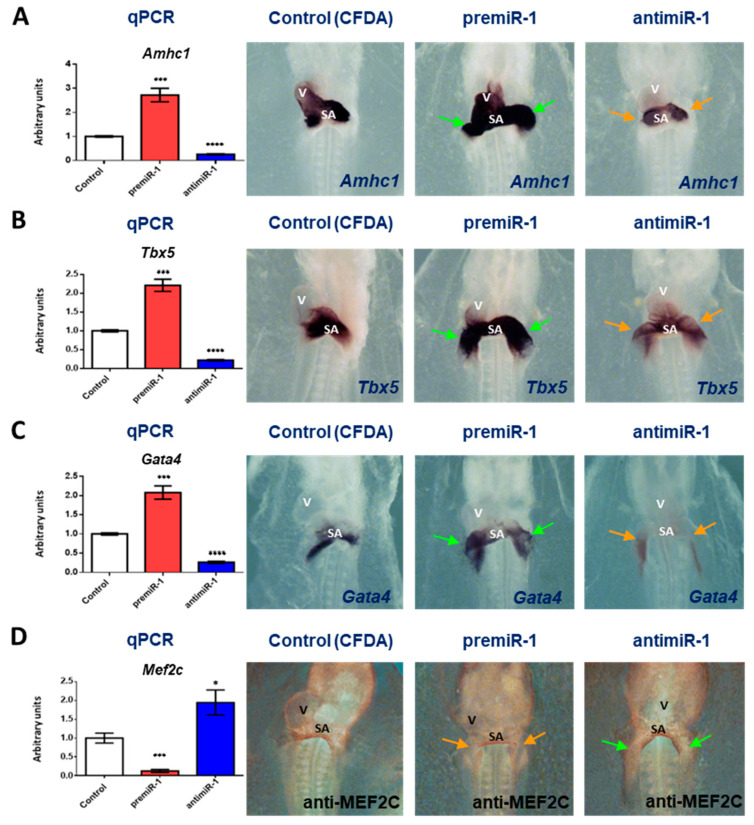
Effects of miR-1 gain- and loss-of-function experiments during the early differentiation of the cardiac sinoatrial region. Whole-mount in situ hybridization (ISH) for *Amhc1 (***A**) *Tbx5* and (**B**) *Gata4* (**C**) and immunohistochemistry (IMH) for MEF2C (**D**). Note the increased and expanded expression of *AMHC1*, *Tbx5* and *Gata4* after premiR-1 treatment (green arrows), accompanied by diminished protein levels of MEF2C (orange arrow). The posterior diminished expressions of *Amhc1*, *Tbx5* and *Gata4* in the heart tube after antimiR-1 treatment are indicated by orange arrows, whereas *Mef2c* was increased (green arrows). The left side illustrates the RT-qPCR of RNA from the dissected cardiac *asa* of embryos microinjected either with CFDA, premiR-1 or antimiR-1. A high level of miR-1 led to increased *Amhc1*, *Tbx5* and *Gata4* transcripts, whereas miR-1 inhibition led to *Mef2c* increased transcripts. V: ventricle; SA: sinoatrial region. Standard deviations are from three independent experiments. Student’s *t*-test: * *p* < 0.05, *** *p* < 0.005, **** *p* < 0.001 with respect to control (CFDA) embryos.

**Figure 3 ijms-25-06608-f003:**
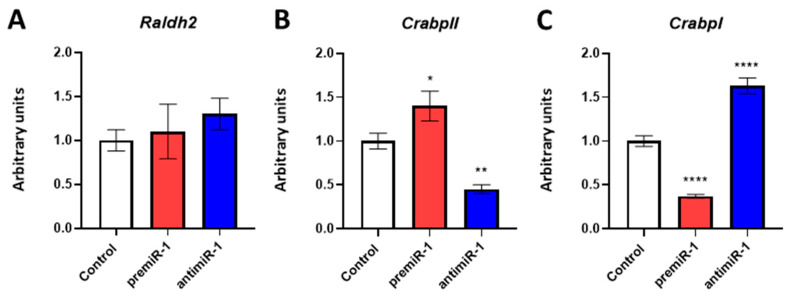
Effects of miR-1 gain- and loss-of-function experiments on RA pathway during posterior differentiation of cardiac tube. RT-qPCR of RNA from the dissected cardiac *asa* of embryos microinjected either with CFDA, premiR-1 or antimiR-1, illustrating *Raldh2* (**A**), *CrabpII* (**B**) and *CrabpI* (**C**). A high level of miR-1 led to an increased expression of *CrabpII* (**B**) and a decreased expression of *CrabpI* (**C**). The opposite effect was observed with anti-miR1 treatment. Note that nonsignificant expression of *Raldh2* (**A**) was observed after miR-1 overexpression or miR-1 inhibitor treatment, as compared to the control (CFDA). Standard deviations are from three independent experiments. Student’s *t*-test: * *p* < 0.05, ** *p* < 0.01, **** *p* < 0.001 with respect to control (CFDA) embryos.

**Figure 4 ijms-25-06608-f004:**
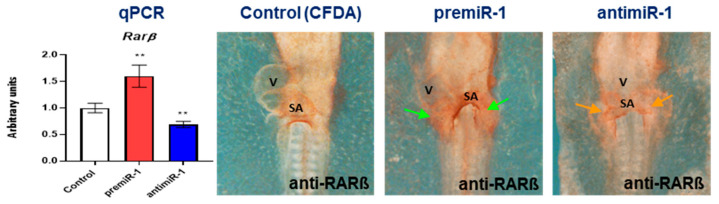
Embryos subjected to *Rarß* study. Whole-mount IMH illustrate RARß markedly increased in the sinoatrial (SA) region after miR-1 administration (green arrows). Note RARß diminished after antimiR-1 administration (orange arrows). RT-qPCR of RNA from the dissected cardiac *asa* (left side) show the levels of *Rarß* transcripts. V: ventricle. Standard deviations are from three independent experiments. Student’s *t*-test: ** *p* < 0.01 with respect to control (CFDA) embryos.

**Figure 5 ijms-25-06608-f005:**
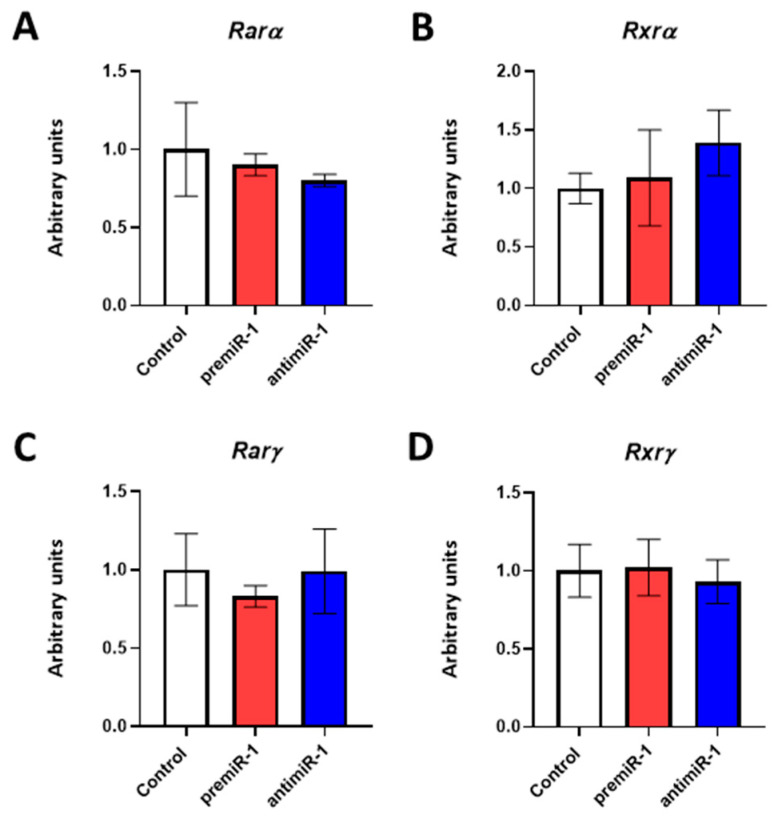
Effects of miR-1 gain- and loss-of-function experiments on RA pathway during posterior differentiation of cardiac tube. RT-qPCR of RNA from the dissected cardiac *asa* of embryos microinjected either with CFDA, premiR-1 or antimiR-1. Note that nonsignificant expression of *Rarα* (**A**), *Rxrα* (**B**), *Rarγ* (**C**) and *Rxrγ* (**D**) were observed after miR-1 overexpression or miR-1 inhibitor treatment as compared to the control (CFDA). Standard deviations are from three independent experiments.

**Figure 6 ijms-25-06608-f006:**
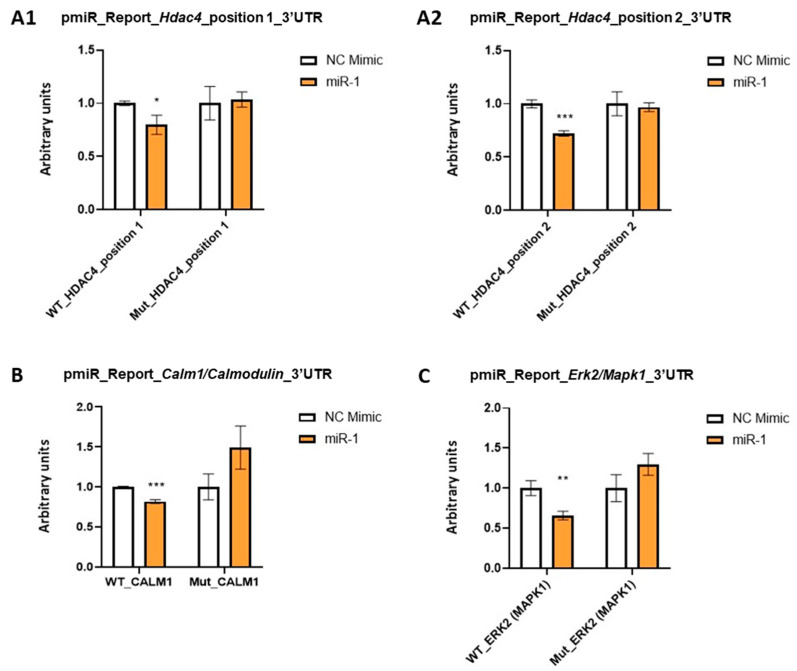
Dual-luciferase activity assay in 3T3 cells cotransfected with the pmiR_Report_Gluc miRNA expression reporter vector containing the wild-type (WT) or mutant (Mut) *Hdac4* (**A1**): position 1; (**A2**): position 2, *Calm1* (**B**) and *Erk2*/*Mapk1* (**C**) 3′UTR fragment with premiR-1. Luciferase activity was compared to non-transfected controls. Each luciferase assay was carried out in triplicate. Student’s *t*-test: * *p* < 0.05, ** *p* < 0.01, *** *p* < 0.005.

**Figure 7 ijms-25-06608-f007:**
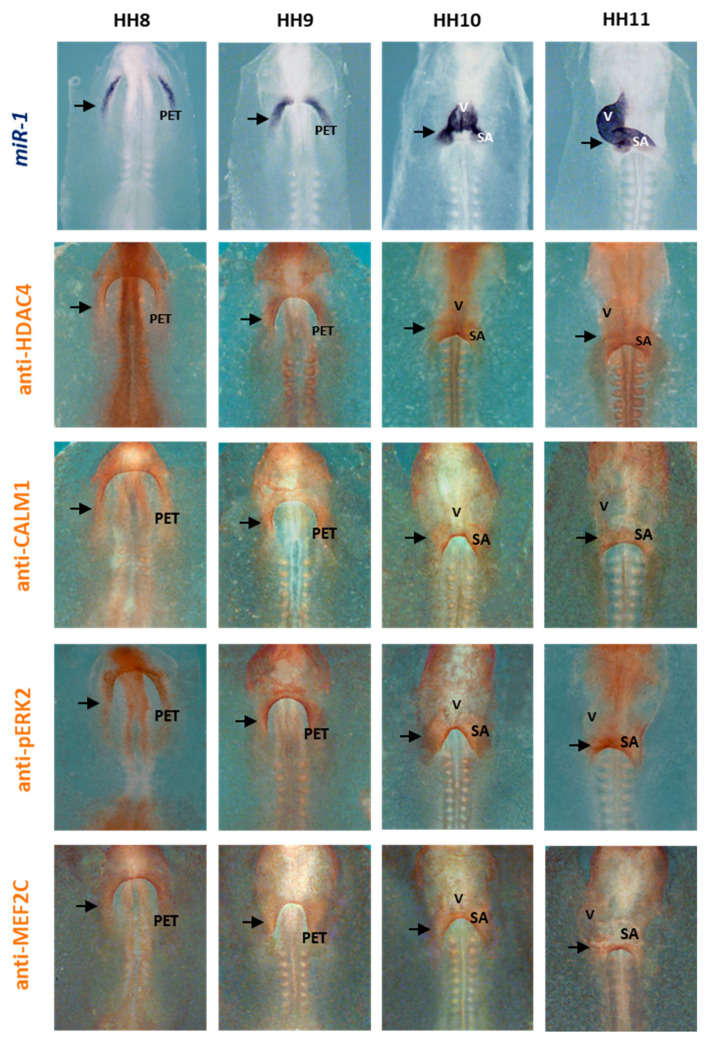
Whole-mount ISH for miR-1 and whole-mount IMH for HDAC4, CALM1, phospho-ERK2 (pERK2) and MEF2C during early chick cardiac development, from HH8 through HH11, in control embryos. Note the miR-1 expression pattern in both primitive endocardial tubes (PET), being observable in the sinoatrial (SA) region and ventricle (V) at later stages. Note the location of HDAC4, CALM1, pERK2 and MEF2C in the PET and, subsequently, in the SA region. Arrows: positive stains.

**Figure 8 ijms-25-06608-f008:**
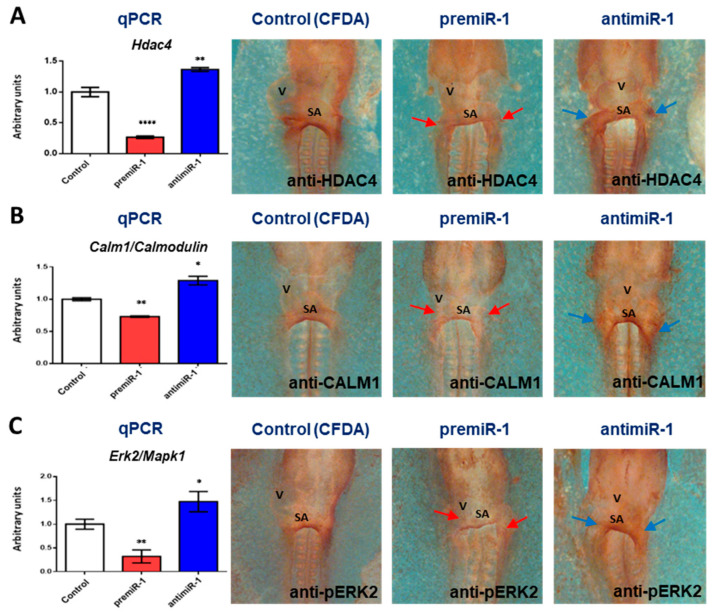
Effects of miR-1 gain- and loss-of-function on target genes *Hdac4*, *Calm1* and *Erk2*/*Mapk1* during posterior differentiation of cardiac tube. Whole-mount IMH reveal that HDAC4 (**A**), CALM1 (**B**) and pERK2 (**C**) were dramatically reduced in the sinoatrial region (red arrows) after premiR-1 administration, whereas they were markedly increased after miR-1 inhibition (blue arrows). RT-qPCR of RNA from dissected cardiac *asa* (left side) in embryos microinjected either with CFDA, premiR-1 or antimiR-1 showed that miR-1 led to decreased *Hdac4*, *Calm1* and *Erk2*/*Mapk1* transcripts, whereas miR-1 inhibition led to increased transcripts. V: ventricle; SA: sinoatrial region. Standard deviations are from three independent experiments. Student’s *t*-test: * *p* < 0.05, ** *p* < 0.01, **** *p* < 0.001 with respect to control (CFDA) embryos.

**Figure 9 ijms-25-06608-f009:**
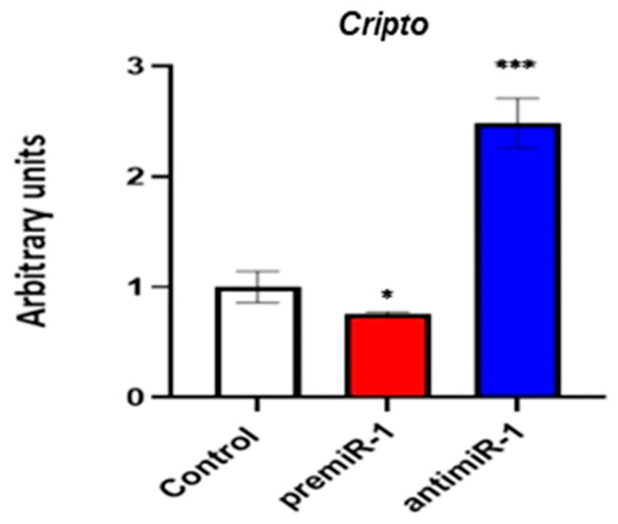
Effects of miR-1 gain- and loss-of-function experiments on *Cripto* during posterior differentiation of cardiac tube. RT-qPCR of RNA from dissected cardiac *asa* of embryos microinjected either with CFDA, premiR-1 or antimiR-1. A high level of miR-1 led to the decreased expression of *Cripto*. The opposite effect was observed with anti-miR1 treatment. Standard deviations are from three independent experiments. Student’s *t*-test: * *p* < 0.05, *** *p* < 0.005 with respect to control (CFDA) embryos.

**Figure 10 ijms-25-06608-f010:**
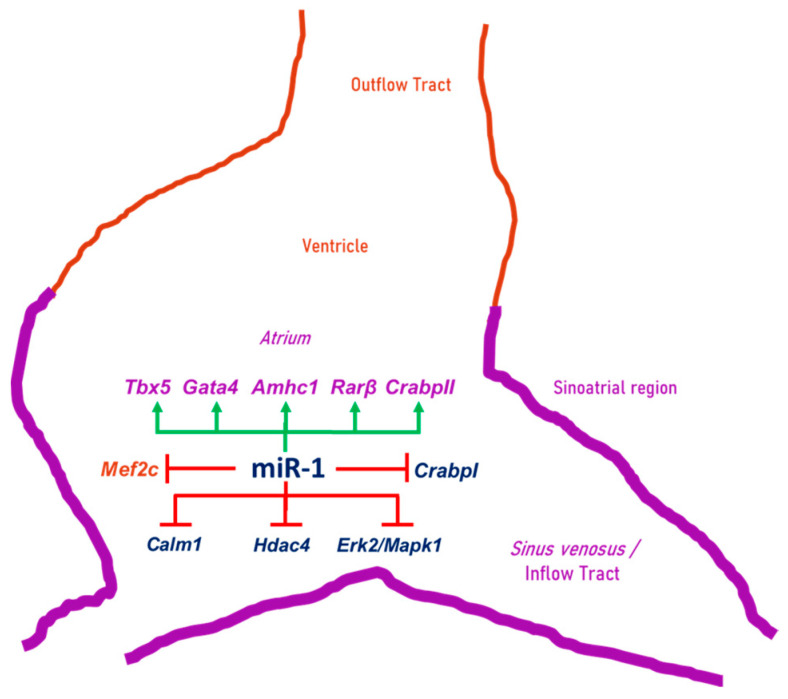
Schematic drawing that summarizes the results of this work. miR-1 induces *Tbx5*, *Gata4*, *Amhc1*, *RARβ* and *CrabpII*, while it suppresses *CrabpI*, *Calm1*, *Hdac4* and *Erk2*/*Mapk1*, as well as *Mef2c*, thus modulating early differentiation of the cardiac sinoatrial region.

**Figure 11 ijms-25-06608-f011:**
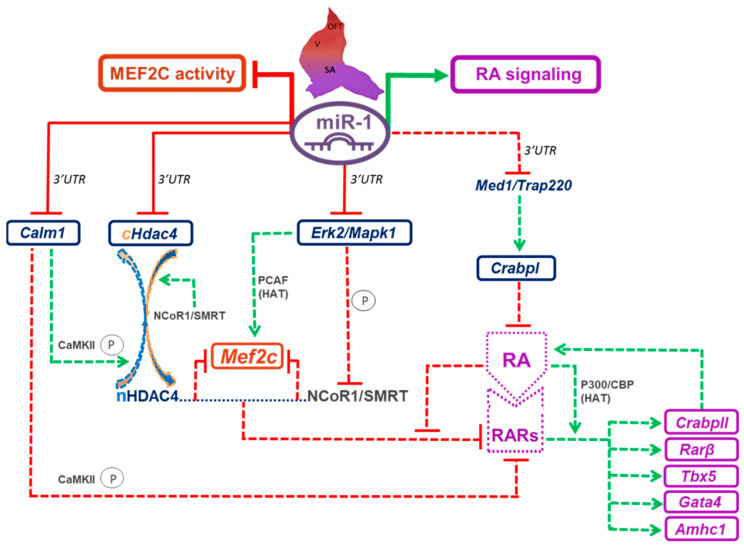
Proposed model illustrating the network of molecular mechanisms governed by miR-1 during early differentiation of the cardiac sinoatrial region. Solid lines indicate our results from this work. Dashed lines correspond to previous studies in different research fields referred in the Section 3. Our model indicates that miR-1 plays a crucial role repressing *Mef2c* through modulation of *Hdac4*, *Calm1* and *Erk2*/*Mapk1*. In these molecular mechanisms, miR-1 modulates opposite effects between *Calm1* and NCoR1/SMRT on nuclear HDAC4 (nHDAC4), thus increasing *Mef2c*’s interaction with nHDAC4, which inhibits *Mef2c* expression. Also, miR-1 suppresses *Erk2*/*Mapk1* and, consequently, diminishes MEF2C activity. Additionally, our model indicates that RA function is modulated by miR-1, thus promoting the expression of RA target genes *Tbx5*, *Gata4*, *Amhc1* and *Rarβ*. Moreover, miR-1 increases *CrabpII* and suppresses *CrabpI*, thus increasing RA activity. The presence of RA disrupts the interaction of nHDAC4-NCoR1/SMRT with RARs, thus allowing transcription of RARs target genes. Also, miR-1 represses *Calm1*’s capability to inhibit RARs’ activity, thus enhancing the expression of RA target genes. In our model, miR-1 modulates the opposite actions between RA and MEF2C, promoting the RA signaling pathway and suppressing MEF2C activity to allow properly assign cells to their cardiac chamber. OFT: outflow tract, V: ventricular region, SA: sinoatrial region, P: phosphorylate.

## Data Availability

The original contributions presented in the study are included in the article/Appendix A, further inquiries can be directed to the corresponding author/s.

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
