# Peer review of "miR-1 as a Key Epigenetic Regulator in Early Differentiation of Cardiac Sinoatrial Region"

_ijms, 2024, doi:10.3390/ijms25126608_

Round 1

Reviewer 1 Report (New Reviewer)

Comments and Suggestions for Authors

The submitted work is a potentially exciting study on the effects of mir-1 microRNA in early avian heart development. However, several aspects should be improved to make the claims of this study robust and vigorous.

1.Authors should specify that mir-1 interacts with the transcript.

2.Authors should unify how they write gene and transcript names following official/recommended gene names: http://birdgenenames.org and pay attention to capital/lowercase and italicized versus roman.

3.Multiple stainings, such as Mef2c, pErk2, and Calmodulin, are not convincing. If authors administer miRNA specifically into the developing heart, staining intensity should stay the same outside the developing heart.

4.Authors should clearly define the nature and number of experimental units and show each unit as an individual point in graphs.

5.When working with less common model organisms like chickens, authors should provide evidence about the antibody specificity used in the studied species.

6.Using bioinformatic methods, authors should predict which transcripts are direct targets of mir-1 and provide data on the conservation of these interactions in humans. 

7.Provide evidence that the used miRNA inhibitor is specific to chicken mir-1.

8.How ISH, IHC, and RT-PCR results were quantified is unclear.

pMIR_REPORT vector encodes firefly luciferase that cannot be measured with Pierce™ Gaussia Luciferase Flash Assay Kit, as the authors claim. 9.Furthermore, the authors need to clarify what the pcLux vector refer

Comments on the Quality of English Language

The submitted work is a potentially exciting study on the effects of mir-1 microRNA in early avian heart development. However, several aspects should be improved to make the claims of this study robust and vigorous.

1.Authors should specify that mir-1 interacts with the transcript.

2.Authors should unify how they write gene and transcript names following official/recommended gene names: http://birdgenenames.org and pay attention to capital/lowercase and italicized versus roman.

3.Multiple stainings, such as Mef2c, pErk2, and Calmodulin, are not convincing. If authors administer miRNA specifically into the developing heart, staining intensity should stay the same outside the developing heart.

4.Authors should clearly define the nature and number of experimental units and show each unit as an individual point in graphs.

5.When working with less common model organisms like chickens, authors should provide evidence about the antibody specificity used in the studied species.

6.Using bioinformatic methods, authors should predict which transcripts are direct targets of mir-1 and provide data on the conservation of these interactions in humans. 

7.Provide evidence that the used miRNA inhibitor is specific to chicken mir-1.

8.How ISH, IHC, and RT-PCR results were quantified is unclear.

pMIR_REPORT vector encodes firefly luciferase that cannot be measured with Pierce™ Gaussia Luciferase Flash Assay Kit, as the authors claim. 9.Furthermore, the authors need to clarify what the pcLux vector refer

Author Response

Reviewer #1

R: The submitted work is a potentially exciting study on the effects of mir-1 microRNA in early avian heart development. However, several aspects should be improved to make the claims of this study robust and vigorous.

A: Dear Reviewer, thank you very much for your valuable feedback. Below you will find point-by-point answer to your queries.

R:1. Authors should specify that mir-1 interacts with the transcript.

A: There are indeed specific interactions between miR-1 and different transcripts, as we demonstrate in the new data provided in Figure 6.

R: 2. Authors should unify how they write gene and transcript names following official/recommended gene names: http://birdgenenames.org and pay attention to capital/lowercase and italicized versus roman.

A: This point has been addressed. Following your suggestion, the corresponding changes have been made.

R: 3. Multiple stainings, such as Mef2c, pErk2, and Calmodulin, are not convincing. If authors administer miRNA specifically into the developing heart, staining intensity should stay the same outside the developing heart.

A: Figures have been enlarged and improved for a better illustration of stainings in this revised version. We have enlarged the image corresponding to the cardiac area (and we have included the images in a better quality and resolution, using TIFF instead JPEG). Also, we have included quantification of signal expression levels and positive area of cardiac sinoatrial region, as Supplementary Figures S1 and S2.

R: 4. Authors should clearly define the nature and number of experimental units and show each unit as an individual point in graphs.

A: According with this indication, we have included new Supplementary Tables S1-S4 in this revised version of the manuscript, including number of embryos (n) subjected to each of the experimental procedures.

R: 5. When working with less common model organisms like chickens, authors should provide evidence about the antibody specificity used in the studied species.

A: The specific characteristics of every antibody is provided by the manufacturer (Predicted reactivity), as referred in Material and Methods (reference code). Additionally, we also provided evidences of the antibody specificity in our immunostained control series as referred to negative controls lacking the primary antibody, which in all cases revealed no staining at all.

R: 6. Using bioinformatic methods, authors should predict which transcripts are direct targets of mir-1 and provide data on the conservation of these interactions in humans.

A: In this sense, we include the new Supplementary Figures S5-S7.

R: 7. Provide evidence that the used miRNA inhibitor is specific to chicken mir-1.

A: In this sense, we included additional data in the new Figure 1: we have performed RT-qPCR analyses, validating the miR-1 gain- and loss-of-function results, increased or decreased significantly miR-1 expression, respectively, as compared to controls.

R: 8. How ISH, IHC, and RT-PCR results were quantified is unclear.

A: To clarify this point we have included the new Supplementary Figures S1 and S2. We have provided additional quantitative analyses of the ISH and IHC stainings, that are included as new Supplementary Figures S1 and S2. Furthermore, detailed description of the quantitation procedures is provided in the new Material and Methods subheading: 4.9 Image analysis. Additionally, RT-qPCR quantitative analyses are detailed in the subheading 4.5: RNA isolation and RT-qPCR.

R: 9. pMIR_REPORT vector encodes firefly luciferase that cannot be measured with Pierce™ Gaussia Luciferase Flash Assay Kit, as the authors claim.

A: This has been clarified modifying 4.7. Luciferase assays and 3T3 transfection in Material and Methods section.

R: 9. Furthermore, the authors need to clarify what the pcLux vector refer.

A: In the same section above mentioned, this “lapsus” has been corrected.

We think that all these above answers improve the Methods description, the presentation of the Results and the Conclusions are supported by the Results.

Reviewer 2 Report (New Reviewer)

Comments and Suggestions for Authors

Dear authors,

congratulations to a very interesting study. Your results are interesting to the scientists in the field of cardiac development and cardiac modeling. Especially figure 9 illustrates the results of the study. Please have a further look at the nomenclature of writing.

Author Response

Reviewer #2

R: Dear authors, congratulations to a very interesting study. Your results are interesting to the scientists in the field of cardiac development and cardiac modeling. Especially figure 9 illustrates the results of the study. Please have a further look at the nomenclature of writing.

A: Very grateful for your exciting comments. Thank you so much. We thank the Reviewer report for finding our paper interesting. Following your suggestion, we have corrected the nomenclature of writing.

Reviewer 3 Report (New Reviewer)

Comments and Suggestions for Authors

The manuscript looks at possible targets of miR-1 in cardiac differentiation in the chicken model.   The approach is interesting but the data was not as clear as it could be and the link to Mef2c signaling was not well developed.

1.  First section of the results before 2.1 should be removed as it is a summary/discussion prior to the presentation of the results themselves.

2. Line 133, it is not clear what is meant by "the effects observed with highlighted RA levels"

3. Figure 1E, the difference in Mef2c levels was not very evident.

4.  Overall the connection to Mef2c signaling was not well developed mechanistically.

5. in the luciferase experiments, the mir-1 sites should be mutated to show that is specific to the predicted sites.

6.  IHC stainings in Figures 6 and 7 were not not clear and the differences between conditions were not evident.

Comments on the Quality of English Language

English was ok but should be proofed.

Author Response

Reviewer #3

R: The manuscript looks at possible targets of miR-1 in cardiac differentiation in the chicken model. The approach is interesting but the data was not as clear as it could be and the link to Mef2c signaling was not well developed.

A: Dear Reviewer, thank you very much for your valuable suggestions. We thank the Reviewer for finding our approach interesting. Based on your suggestion, we have improved the explanation of the link to Mef2c signaling. Likewise, we have addressed all the points mentioned.

R: 1. First section of the results before 2.1 should be removed as it is a summary/discussion prior to the presentation of the results themselves.

A: We have deleted the aforementioned paragraph.

R: 2. Line 133, it is not clear what is meant by "the effects observed with highlighted RA levels"

A: To clarify this aspect, we have modified this paragraph in the main text. Now … thus revealing that miR-1 mimics the effects observed after RA administration.

R: 3. Figure 1E, the difference in Mef2c levels was not very evident.

A: To address this point, we have modified Figure 2 to improve the quality of the images. We have enlarged the image corresponding to the cardiac area (and we have included the images in a better quality and resolution, using TIFF instead JPEG). Moreover, we have included the new Supplementary Figure S1, including quantification of signal expression levels and positive area in cardiac sinoatrial region.

R: 4. Overall the connection to Mef2c signaling was not well developed mechanistically.

A: In this sense we have included a new experiment. We have analyzed the effect of miR-1 on Cripto, a direct transcriptional target of MEF2C. The results of these additional experiments have been included and discussed in this new version.

R: 5. in the luciferase experiments, the mir-1 sites should be mutated to show that is specific to the predicted sites.

A: Following this indication, we have included additional data in this version, as illustrated in the new Figure 6, by means of dual-luciferase activity assay in 3T3 cells co-transfected with the pmiR_Report_Gluc miRNA expression reporter vector containing the wild-type or mutant Hdac4, Calm1 or Erk2/Mapk1 3’UTR fragment with premiR-1.

R: 6. IHC stainings in Figures 6 and 7 were not not clear and the differences between conditions were not evident.

A: We have improved both Figures 6 and 7, which are 7 and 8 in this revised version. We have enlarged the image corresponding to the cardiac area (and we have included the images in a better quality and resolution, using TIFF instead JPEG). Additionally, Supplementary Figure S2, includes quantification of signal expression levels and positive area in cardiac sinoatrial region.

R: English was ok but should be proofed.

A: The English edition has been revised.

We think that all these above answers improve the presentation of the Results and the Conclusions are supported by the Results.

Round 2

Reviewer 3 Report (New Reviewer)

Comments and Suggestions for Authors

My concerns regarding image quality for immunostainings remain.  Some of the described changes in expression are not apparent.   Validation with another approach such as qPCR is needed to strengthen their conclusions.

Author Response

Reviewer #3

R: My concerns regarding image quality for immunostainings remain.  Some of the described changes in expression are not apparent.

A: Dear Reviewer, thank you very much for your indications. We have improved the immunostaining figures. In addition, we have uploaded a Zip with the Original Figures in high resolution in a separate file.

R: Validation with another approach such as qPCR is needed to strengthen their conclusions.

A: Each Figure of our article (and Supplementary Materials) corresponding to our experimental assays is strongly supported by the corresponding qPCR analyses, illustrated alongside each image.

Round 3

Reviewer 3 Report (New Reviewer)

Comments and Suggestions for Authors

No further changes

This manuscript is a resubmission of an earlier submission. The following is a list of the peer review reports and author responses from that submission.

Round 1

Reviewer 1 Report

Comments and Suggestions for Authors

In this study, the investigators injected either premiR-1 or antimiR-1 into the primitive endocardial tubes (PET) of developing chick embryos. The embryos were later fixed and stained for histology analysis, analyzed by PCR or for RNA expression of cardiac markers, or luciferase assays for localization of ventricular markers. They found that miR-1 increased expression of sinoatrial transcription factors and downregulated the Mef2c marker of ventricular formation. Additionally, they found increases in SA development in embryos injected with miR-1, and the opposite effects with antimiR-1. Further analysis suggests this acts through the RA signaling and ERK/MAPK signaling pathways.

I have only the following minor comments:

1) The authors should explain the premiR structure and reasoning for those not familiar with microRNA.

2) I suggest providing a quantitative measurement (area or percentage) of structural SA and ventricular regions shown in representative stains in figure 1 and supplemental figure 1.

Author Response

Comments and Suggestions for Authors

In this study, the investigators injected either premiR-1 or antimiR-1 into the primitive endocardial tubes (PET) of developing chick embryos. The embryos were later fixed and stained for histology analysis, analyzed by PCR or for RNA expression of cardiac markers, or luciferase assays for localization of ventricular markers. They found that miR-1 increased expression of sinoatrial transcription factors and downregulated the Mef2c marker of ventricular formation. Additionally, they found increases in SA development in embryos injected with miR-1, and the opposite effects with antimiR-1. Further analysis suggests this acts through the RA signaling and ERK/MAPK signaling pathways.

A: Thank you very much for your positive and constructive criticisms.

Minor comments:

  • 1) The authors should explain the premiR structure and reasoning for those not familiar with microRNA.

A: According to the Reviewer´s indication, we have included in Material and Methods explanation about the premiR structure, and also adding two more References. (Lines 443-447).

  • 2) I suggest providing a quantitative measurement (area or percentage) of structural SA and ventricular regions shown in representative stains in figure 1 and supplemental figure 1.

A: Following the Reviewer´s suggestion, we provide additional information about the area percentage of structural SA and ventricular regions, adding to Figure Supplementary S1 (B) the data obtained from those embryos subjected to miR-1 experiments.

Reviewer 2 Report

Comments and Suggestions for Authors

This study investigates the role of miR-1, a significant microRNA in cardiac development, in regulating cardiac chamber differentiation via specific signaling pathways. Through microinjections into chick embryos' cardiac precursors, the research explores miR-1's impact on gene expression, focusing on atrial genes like Tbx5, Gata4, and AMHC1, as well as Mef2c. The findings suggest that miR-1 influences retinoic acid signaling by modulating CRABPII, RARß, and CRABPI and interacts with HDAC4, Calmodulin, and Erk2/MAPK1, essential factors in Mef2c regulation. AntimiR-1 administration produced contrasting results. Overall, this work elucidates miR-1's role as an epigenetic factor in orchestrating actions involved in assigning cells as sinoatrial precursors, offering insights for therapeutic applications in cardiac regeneration and repair. However, the novelty of this study has been low.

Major:

1.     To further demonstrate the specificity of miR-1 in regulating these genes, except the anti-miR-1, the mutant miR-1 that lacks binding sequence to these genes should be examined in the microinjection assays.

2.     Did the authors examine the expression of phosphor-Erk2 in the microinjection assays? This would be more important than total Erk2 expression, as activated Erk2 promotes signal transduction.

Minor:

1.     It would be better to use arrows to point out the positive stains in Figure 5.

2.     The findings that miR-1 works as an epigenetic regulator for some genes (i.e., Tbx5, Hdac4) have been well established. It would be essential for the authors to emphasize the novelty of the current study.

3.     In the Abstract, Line 22, why/how did the authors think Tbx5 and Gata4 are “atrial genes”?

4.     The first paragraph of the “Results” section (Lines 93-105) should be moved and consolidated into the “Introduction” or “Discussion” section. 

Author Response

Comments and Suggestions for Authors

This study investigates the role of miR-1, a significant microRNA in cardiac development, in regulating cardiac chamber differentiation via specific signaling pathways. Through microinjections into chick embryos' cardiac precursors, the research explores miR-1's impact on gene expression, focusing on atrial genes like Tbx5, Gata4, and AMHC1, as well as Mef2c. The findings suggest that miR-1 influences retinoic acid signaling by modulating CRABPII, RARß, and CRABPI and interacts with HDAC4, Calmodulin, and Erk2/MAPK1, essential factors in Mef2c regulation. AntimiR-1 administration produced contrasting results. Overall, this work elucidates miR-1's role as an epigenetic factor in orchestrating actions involved in assigning cells as sinoatrial precursors, offering insights for therapeutic applications in cardiac regeneration and repair. However, the novelty of this study has been low.

A: Thank you very much for your positive and constructive criticisms. Following the indications, we have highlighted the originality and novelty of this study, mainly in the Discussion section.

Major comments:

  • 1. To further demonstrate the specificity of miR-1 in regulating these genes, except the anti-miR-1, the mutant miR-1 that lacks binding sequence to these genes should be examined in the microinjection assays.

A: The specificity of the miR-1 action could indeed theoretically be validated by generating mR-1 molecules with mutant seed sequences. However, to our knowledge would imply mutating the seed sequences of the pre-miR-1 molecules. Endogenous pre-miRNA-1 mimics are commercially available, but not mutated mimics. We have nonetheless validated the specificity of miR-1 mimics by using commercially available scrambled miRNA mimics in other experimental settings, yielding in all cases no significant differences of target expression (Daimi et al., 2015). Therefore, we have not repeated these experiments in this study.

[Daimi H, Lozano-Velasco E, Haj Khelil A, Chibani JB, Barana A, Amorós I, González de la Fuente M, Caballero R, Aranega A, Franco D. Regulation of SCN5A by microRNAs: miR-219 modulates SCN5A transcript expression and the effects of flecainide intoxication in mice. Heart Rhythm. 2015 Jun;12(6):1333-42. doi: 10.1016/j.hrthm.2015.02.018. Epub 2015 Feb 19. PMID: 25701775]

  • 2. Did the authors examine the expression of phosphor-Erk2 in the microinjection assays? This would be more important than total Erk2 expression, as activated Erk2 promotes signal transduction.

A: In reference to the Reviewer´s question, the answer is “yes”. Please note that in Material and Methods Section, 4.4, we wrote: “we are using polyclonal antibody phospho-Erk1/Erk2 (1:20, Invitrogen 44-680G)”. Line 471. Also, it has been now specified in Text and Figure legends corresponding to Figures 5 and 6.

Minor comments:

  • 1. It would be better to use arrows to point out the positive stains in Figure 5.

A: According to the Reviewer´s suggestion, we introduced some arrows to point out the positive stains in Figure 5.

  • 2. The findings that miR-1 works as an epigenetic regulator for some genes (i.e., Tbx5, Hdac4) have been well established. It would be essential for the authors to emphasize the novelty of the current study.

A: Following the Reviewer´s suggestion, we emphasize the novelty of the current study in the Discussion section.

  • 3. In the Abstract, Line 22, why/how did the authors think Tbx5 and Gata4 are “atrial genes”?

A: According to the Reviewer´s indication, in this sense, we have modified the paragraph.

  • 4. The first paragraph of the “Results” section (Lines 93-105) should be moved and consolidated into the “Introduction” or “Discussion” section.

A: Following the Reviewer´s indication, the first paragraph of the “Results” section have been moved and consolidated into the “Discussion” section. Lines 274-278.

Round 2

Reviewer 2 Report

Comments and Suggestions for Authors

The authors have addressed most of my concerns. I have no further questions.